# Microbial Community Structure Driven by a Volcanic Gradient in Glaciers of the Antarctic Archipelago South Shetland

**DOI:** 10.3390/microorganisms9020392

**Published:** 2021-02-14

**Authors:** Eva García-Lopez, Sandra Serrano, Miguel Angel Calvo, Sonia Peña Perez, Silvia Sanchez-Casanova, Laura García-Descalzo, Cristina Cid

**Affiliations:** Molecular Evolution Department, Centro de Astrobiologia (CSIC-INTA), Torrejón de Ardoz, 28850 Madrid, Spain or garciale@cab.inta-csic.es (E.G.-L.); sandra.se.ho@gmail.com (S.S.); miguel.calvo53@gmail.com (M.A.C.); soniapenaperez@gmail.com (S.P.P.); silviasanchez1@hotmail.com (S.S.-C.); laura_garcia_descalzo@hotmail.com (L.G.-D.)

**Keywords:** cryosphere microbiome, 16S/18S rRNA high throughput sequencing, glacier, biodiversity, ecology, Antarctica, biogeochemistry, astrobiology

## Abstract

It has been demonstrated that the englacial ecosystem in volcanic environments is inhabited by active bacteria. To know whether this result could be extrapolated to other Antarctic glaciers and to study the populations of microeukaryotes in addition to those of bacteria, a study was performed using ice samples from eight glaciers in the South Shetland archipelago. The identification of microbial communities of bacteria and microeukaryotes using 16S rRNA and 18S rRNA high throughput sequencing showed a great diversity when compared with microbiomes of other Antarctic glaciers or frozen deserts. Even the composition of the microbial communities identified in the glaciers from the same island was different, which may be due to the isolation of microbial clusters within the ice. A gradient in the abundance and diversity of the microbial communities from the volcano (west to the east) was observed. Additionally, a significant correlation was found between the chemical conditions of the ice samples and the composition of the prokaryotic populations inhabiting them along the volcanic gradient. The bacteria that participate in the sulfur cycle were those that best fit this trend. Furthermore, on the eastern island, a clear influence of human contamination was observed on the glacier microbiome.

## 1. Introduction

The cryosphere is experiencing drastic changes due to its high sensitivity to global warming. The consequences of these changes are shown at many levels. For example, effects of climatic change on vascular plants and lichens have been recently described [1,2]. Taking into account that microorganisms are the base of trophic webs, their changes affect the entire ecosystem. Both abiotic (i.e., natural forces, volcanic eruptions) and biotic factors (i.e., human presence, animal and plant populations, etc.) may positively or negatively feedback these effects. The absence of detailed knowledge of the prokaryotic diversity in poles is a major omission, taking into account the prokaryotes’ numerical abundance and their importance in biogeochemical cycles. The South Shetland archipelago is an excellent example of a polar ecosystem to expand our knowledge of glacier microbiomes and how environmental changes affect them.

Glaciers are a fundamental part of the cryosphere. Recent applications of molecular genetics to microbial communities of Antarctic glaciers have rejected the belief that Antarctic glaciers host extremely limited microbial diversity. On the contrary, they not only contain a great diversity of microorganisms, but also new mechanisms of adaptation to the environment since selection acts in them with a special intensity. These features make these ecosystems unique and interesting for the study and protection of biological heritage. It is important to investigate how microbial patterns are altered and how these changes in turn affect the climate because microorganisms are essential components of food chains and are dominant primary producers. Moreover, these microorganisms can provide important data on the thermodynamic limits of life, which may be applicable in other disciplines such as astrobiology [3].

Among glaciers that have been drilled to study microbial populations, some examples can be cited in Antarctica [4,5], in the Arctic region in Greenland [6,7] and Alaska [8], and in the Tibetan plateau [9]. Three ecological units have been defined as microbial habitats on, within, and beneath glaciers: the supraglacial, englacial, and subglacial ecosystems [10,11]. 

The upper stratum (snow, surface ice, and cryoconite holes), well oxygenated and illuminated by the sun, is predominantly populated by photoautotrophic microorganisms like cyanobacteria and microalgae. When it snows, snowflakes draw bacteria from the air. These upper microorganisms colonize the inner part of the glacier through small channels. In the inner layers, in which dust particles and sand are accumulated, the population density increases. On icy surfaces, small cavities are formed on ice as a result of the deposition of cryoconite, i.e., a mixture of organic and inorganic dark particles which absorb sunlight and cause ice melting. These cryoconite holes have been widely studied because of their highly diverse and active bacterial populations [12,13]. 

The subglacial ecosystem is generated as a result of the friction of the ice on the rocks, creating thinner material. These fragments contain minerals and organic sedimentary carbon which combined with subglacial water creates niches optimal for microbial life. Over time, the liquid water beneath the layers of ice can form streams and lakes. Few studies have sampled subglacial environments given the challenges associated with the access to these places, but several species of archaea, bacteria, and fungi have been found. Most are chemoautotrophs which feed of minerals from rocks and soil. 

Between these two ecosystems there is a hidden englacial ecosystem, much less studied. It had always been considered uninhabited or inhabited by dormant microorganisms dragged from the upper layers. Recently we have demonstrated that the englacial ecosystem is populated by microorganisms that are also metabolically active [14]. It has been shown that, since the mobility of microorganisms within the ice is nearly impossible, there is communication between microorganisms through molecules that act as regulators of response, production of antibiotics, control of oxidative stress, and reaction to chemical substances [14]. 

However, these results have been obtained in the glacier of a volcanic island, which makes us question if these characteristics can be extrapolated to other glaciers. To verify this, a study has been done in a transect along eight glaciers, starting from the volcano in Deception Island and heading east to the archipelago South Shetland (Figure 1). 

Volcanic activity is one of the natural phenomena with the highest impact on the diverse Earth’s system constituents. The area affected by an eruption can reach up to tens of thousands of km^2^ [15]. Some of the most relevant consequences include climate effects due to the injection of solid particles and gases into the stratosphere, changes in the temperature and composition of water sources, variations of glaciers due to the deposition of the volcanic material and instantaneous destruction of fauna and flora with the corresponding impact on the ecosystems. In Antarctica, some volcanoes have experienced activity recently, Deception Island being among the most active ones, with various eruptions in the last centuries. Currently, as a result of the successive volcanic explosions that took place on Deception Island in 1967, 1969, and 1970, volcanic remains of lava and other pyroclastic materials are deposited into the glacial ice, forming a gradient from Deception Island to the rest of the archipelago [16]. Volcanic ash deposits are extremely short-lived due to their immediate reworking by water and wind. However, in Antarctica, these materials are embedded in glacial ice and transferred to the environment over the years. These volcanic remains significantly modulate the composition of the glacier ice and may have a dominant role in shaping microbial communities. 

Other factors that can affect microbial populations are animal and human presence. Historically, human contamination in Antarctica has occurred because of explorers, fishers, whalers and more recently, scientific researchers, station support personnel, and commercial and private tourists [17]. It has been reported that human sewage pollution of aquatic environments can increase nutrient load, change pH, temperature, and turbidity, lower dissolved oxygen availability, and contaminate the environment with diverse microorganisms [17,18]. All these factors can alter the composition of local microbial populations.

The specific aims of this study were to test the hypothesis that a gradient of temperature and mineral concentration in glacial ice would cause significant shifts in glacial microbial community composition and their metabolic capacity, taking into account that physicochemical conditions may have the dominant role in shaping microbial populations. To examine this hypothesis, the structure and diversity of glacier bacterial and microeukaryotic communities from eight representative glacier ice samples collected in four South Shetland Islands were analyzed using a molecular (high throughput rRNA gene sequencing)-based approach. Several essential questions were also addressed: (i) Are these glaciers influenced by the Deception Island volcano? (ii) Does this influence diminish with distance? (iii) Are there other factors that influence the diversity of microbial populations (i.e., human or animal pressure)? Together, these studies aimed to improve our understanding of the diversity and possible origins, functions, and trophic interactions of glacier microbiomes.

## 2. Materials and Methods

### 2.1. Sites and Sample Collection

Glacial ice samples were collected at eight sites on the Antarctic Archipelago South Shetland glacier fronts to have access to the englacial zone. Samples were named: Rojo (ROJ) and Macaroni (MAC) glaciers in Deception Island, Johnson (JOH) and Hurd (HUR) glaciers in Livingston Island, Quito (QUI) and Traub (TRA) glaciers in Greenwich Island, and Ecology (ECO) and Machu Pichu (MPI) glaciers in King George Island, in February 2015 (Figure 1). 

GPS coordinates of the sampling points are detailed in Table 1. Ice samples were obtained by removing 20–30 cm of surface debris and extracted by drilling with a Mark II Kovacs Core System. One horizontal ice core of 9 cm x 1 m was extracted at each glacier. Only the innermost end of the horizontal core (60 cm) was considered for the analysis (20 cm for each sample approximately). 

Three sampling replicates were collected from each glacier. Samples were wrapped in sterile plastic bags and stored at −20 °C until analyzing in the laboratory at the Centre for Astrobiology, Madrid, Spain. Samples were decontaminated following the methods described in previous studies [14,19]. Each sample was thawed in sterile containers at 4 °C (1000 mL), and separately used for all the analysis. All procedures were performed by using bleach-sterilized work areas, a UV-irradiated laminar flow hood, ethanol-sterilized tools, and sterilized gloves. To control for laboratory contamination, 1 liter of MilliQ rinse water was subjected to equal analytical procedures.

### 2.2. Chemical Analysis of Meltwater

Meltwater chemistry was analyzed to examine changes along the volcanic gradient. Analyses included in situ pH, temperature, and chemical composition of ice samples. Basic measurements of the physical and chemical parameters of meltwater were made with a temperature-calibrated pH and conductivity meter (WTW, Weilheim, Germany). Assays for NH_4_^+^, NO^2−^, NO^3−^, total dissolved nitrogen (TDN), soluble reactive phosphorus (SRP) and dissolved organic carbon (DOC) from each sample were performed as described elsewhere [20] by ion chromatography in an 861 Advance Compact IC system (Metrohm AG, Herisau, Switzerland). The measurements of major, minor, and trace elements (Ag, Al, As, B, Ba, Be, Bi, Br, C, Ca, Cd, Cl, Co, Cr, Cu, Fe, Ga, Ge, K, Li, Mg, Mn, Mo, Na, Ni, P, Pb, Rb, S, Sc, Se, Sr, Ti, V, Y, Zn, and Zr) (Appendix A) were performed by inductively coupled plasma-mass spectrometry (ICP-MS) on a PerkinElmer ELAN9000 ICP-MS quadrupole spectrometer as in [14]. The samples were introduced into the ICP-MS via a RytonTM cross-flow nebulizer (PerkinElmer, Waltham, MA, USA), Scott spray chamber (PerkinElmer), and Cetac ASX-510 autosampler (Omaha, NE, USA). 

### 2.3. DNA Extraction and Sequencing

Approximately 500 mL of each frozen sample was melted at 4 °C and filtered through a 0.22 µm filter (Millipore). Extraction procedures were identical for all samples. DNA concentration was determined using a Nanodrop 2000p. Then, Illumina sequencing was applied to each sampling replicate. Purified DNA was quantified and 1ng of input DNA was used in a first PCR of 20 cycles with Q5^®^ Hot Start High-Fidelity DNA Polymerase (New England Biolabs) in the presence of 100nM primers of the regions V3 and V4 of the 16S rRNA gene, or of the region V4 and V5 region of the 18S rRNA gene as previously reported [21,22]. Primers sequences were: V3-V4 (Forward ACACTGACGACATGG TTCTACACCTACGGGNGGCWGCAG, Reverse TACGGTACGACTTGGTCTGTACHV GGGTATCTAATCC); V4-V5 (Forward GCCCAVCYGCGGTAAY, Reverse CCGTCAATTHCTTYAART). After the first PCR, a second PCR of 15 cycles was carried out with Q5^®^ Hot Start High-Fidelity DNA Polymerase (New England Biolabs) in the presence of 400 nM of primers Forward AATGTACGCGACCACCGAGATCTACACTGACGACATGGTTCTACA and Reverse CAAGCAGAAGACGGCATACGAGAT [10 nucleotides barcode] TACGGTAGCAGAGACTTGGTCT of the Access Array Barcode Library for Illumina Sequencers (Fluidigm). Amplicons were validated and quantified by Bioanalyzer and an equimolecular pool was purified using AMPure beads and titrated by quantitative PCR using the Kapa-SYBR FAST qPCR kit for LightCycler 480 and a reference standard for quantification. The pool of amplicons were denatured prior to being seeded on a flowcell at a density of 10pM, where clusters were formed and sequenced using a MiSeq Reagent Kit v3, in a 2 × 300 pair-end sequencing run on a MiSeq sequencer to obtain 100,000 reads per sample approximately.

Sequencing data were analyzed with UCHIME [23], to identify and remove chimeric reads, and classified to eliminate those that could be considered contaminants. Sequence reads were grouped according to their taxonomic classification using the BaseSpace platform and the Greengenes 16S rRNA database. In general, this strategy allowed the classification of sequences into taxonomic categories that were never lower than genus rank. The correspondence of operational taxonomic units (OTUs) to species was assessed considering a 97% identity threshold. Analytic Rarefaction 1.3 software (https://strata.uga.edu/software) (accessed on 22 November 2020) was used to perform rarefaction analysis [24,25]. It showed that, at 3% sequence divergence, rarefaction curves nearly reached saturation, indicating that samples contained almost all the diversity at this genetic distance (Appendix A).

### 2.4. Statistical Analysis

Statistical differences in the number of clones and the number of OTUs were studied by the ANOVA test using GraphPad Prism 6.0 software. Data of OTUs and clones are media values of three sampling replicates. Effects of environmental variables on the community composition were investigated by a combination of analysis (DCA, detrended correspondence analysis and CCA, canonical correspondence analysis) developed with CANOCO 5 software [26]. For statistical analysis, Monte Carlo permutation tests with 500 permutations were used.

### 2.5. Nucleotide Sequence Accession Numbers

Sequences obtained from 16S rRNA sequencing were deposited in NCBI Short Read Archive (SRA) (accession numbers SAMN14863991, SAMN14868361, SAMN14868400, SAMN14868414, SAMN14868429, SAMN14868542, SAMN14868580, and SAMN14868598). Sequences obtained from 18S rRNA sequencing were deposited in NCBI SRA (accession numbers SAMN14868636, SAMN14868654, SAMN14868683, SAMN14868702, SAMN14868717, SAMN14868740, SAMN14868844, and SAMN14868902).

## 3. Results and Discussion

### 3.1. General Characteristics of the Ice Samples and Chemical Properties

The South Shetland glaciers studied encompassed a high degree of physicochemical heterogeneity, although the pH values were similar in all the sampling points (Table 2). Temperature measurements in sampling sites showed a significant increase from west to east along the archipelago. 

Among nutrients, NH_4_^+^, NO_2_^−^, NO_3_^−^, TDN, SRP, and DOC (Table 2), the highest concentrations were generally found on King George Island. If only the rest of the islands were considered, the highest concentrations were observed on Deception Island. These organic compounds could come from sewage pollution [17] on the most populated island, King George. In addition, these islands are the most inhabited by seabirds and have a significant population of penguins. The high concentrations of nitrogen compounds observed could also come from the ornitogenic soils that surround the glaciers [27].

Ash and pyroclastic material emitted by volcanoes provide some chemical elements that microorganisms can also use as nutrients (i.e., C, Ca, and Fe). In addition, potentially toxic trace elements (e.g., Al, As, Cu, F, Mo, Ni, Pb, and Zn) are released into the environment. Chemical elements in meltwater were quantified and represented in Appendix A. According to several reports, some of these elements are more involved in the geochemical impact of volcanic ash than others because of their differential mobilization in water after the volcanic eruption [15,28]. It has also been asserted that there are notable variations in the concentrations of the elements involved from volcano to volcano and even between eruptions of the same volcano [15]. Generally, a concentration gradient was observed in these elements, with maximum values in the most extreme islands. From Decepcion Island a descending gradient of elements of volcanic origin (C, Ca, Cd, Fe, S, etc.) was observed. From the eastern islands, there was a westward gradient of other elements (As, Br, Cl, Na, etc.). The observed high values of Cl and Na could come from seawater in glaciers that are most coastal or most exposed to the sea waves. 

### 3.2. 16S and 18S rRNA Gene Clone Libraries

The V3 and V4 regions of the 16S rRNA gene were sequenced by Illumina MiSeq (Appendix A). A total of 602,053 reads were obtained which belonged to 710 OTUs spanning 28 phyla. Microorganisms mainly corresponded to the phyla Firmicutes (29%), Proteobacteria (20%), and Bacteroidetes (19%) (Figure 2A). Some phyla such as Tenericutes, Chrysiogenetes, Chlorobi, and Fusobacteria were scarce. The glaciers with the highest number of different OTUs were MAC and JOH. Some of the identified OTUs such as *Polaribacter*, *Polaromonas*, *Psychrobacter*, *Psychromonas*, and *Cryobacterium* have been described as being typical Antarctic psychrophilic and psychrotolerant bacteria. On the contrary, microorganisms as *Thermobaculum*, *Thermovenabulum*, *Thermoanaerobacterium*, *Thermodesulfovibrio*, *Thermomonas*, and *Caldanaerobacter* were thermophilic or hyperthermophilic. Furthermore, several genera of cyanobacteria as *Chroococcidiopsis* capable of endolithic colonization in volcanic rocks were found [29]. 

Regarding the research of microeukaryotes (Appendix A), amplification and sequencing of the V9 region of the 18S rRNA gene rendered a total of 1,386,341 reads from 1248 OTUs and 20 phyla (Figure 2B). The most frequent phyla were Nucletmycea (42%), Rhizaria (23%), and Chloroplastida (18%). Nucletmycea, that is, fungi had been extensively described in Antarctic soils [30], but they are not found as frequently in glacial ice [31]. We also found several genera of typical endolithic algae belonging to the class Trebouxiophyceae, widespread in the Antarctic Dry Valleys [32]. The most abundant bacterial and eukaryotic OTU in each sampling point has been represented in Appendix A.

### 3.3. Differences in Community Structure among Glaciers

In the studied South Shetland glaciers, a great diversity was observed when compared with other Antarctic glaciers [33] or with the microbiomes of the frozen deserts in the McMurdo Dry Valleys [34]. In these South Shetland glaciers, the communities of microorganisms were isolated from small isolated clusters within the ice. Even the composition of the prokaryotic populations identified in the glaciers from the same island was different (Figure 2). 

In general, the number of 16S rRNA sequences was higher in western glaciers than in eastern glaciers. With the number of 18S rRNA sequences, the opposite trend was observed (Appendix A). Furthermore, the westernmost glaciers in the South Shetland archipelago were inhabited by psychrophilic microorganisms, but also contained bacteria characteristic of volcanic regions such as *Sulfobacillus yellowstonensis* [35], the most abundant bacteria in ROJ samples (Appendix A). These results are probably influenced by the special conditions of Deception Island, an active volcano where glacial ice contains elevated concentrations of certain elements diffusing from geothermal sources. This suggests that active geothermal processes may serve as an additional source of trace elements, or that local physicochemical parameters promote an increase in bioavailability at Deception Island forming a special microbial community. Its fluctuating conditions include temperature gradients, oxygen levels, and multiple forms of chemical energy, such as methane, hydrogen, and hydrogen sulfide. 

In the central islands, psychrophilic microorganisms were more abundant (*Polaribacter*, *Polaromonas*, and *Psychrobacter*) (Appendix A). The eastern islands are those that suffer the greatest human pressure and their glaciers contain microorganisms typical of human (Enterobacteriaceae, *Helicobacter*) or animal contamination (i.e., the microeukaryote *Cryptosporidium*) (Appendix A) [18,36]. Appendix A represents the abundance of some microbial species that could be related to human and animal contamination. An increasing number of reports have shown the introduction and spread of non-native plants [37], shifts of bird and seal breeding areas, and decreases in both bird and seal populations [38]. However, few analyses have been reported at a microbial level. 

Some investigations have demonstrated that the composition of glacier microbiomes depends on the proximity to the sea [21]. In several Arctic coastal glaciers, the genus *Hymenobacter* has been described as a possible sentinel for bacterial transport between glaciers and their downstream seawaters. In these South Shetland glaciers, *Hymenobacter* was also identified in coastal glacier samples. Other typical marine microorganisms found in these glaciers were *Marinobacter* [39], *Polaribacter*, and *Roseospira* [40,41]. Examples of marine microeukaryotes identified in glacier ice include choanoflagellates, alveolates such as *Trichototaxis marina* and *Caudiholosticha marina*, and various types of stramenopiles [42,43].

### 3.4. Distribution of Taxa along the Volcanic Gradient

To study the distribution of microorganisms across all samples, a DCA was carried out taking into account the relative abundances of the microbial phyla. This analysis clustered microorganisms into two groups of bacteria and two groups of microeukaryotes. Therefore, bacteria and microeukaryotes associate in separate groups (Figure 3). 

To check if there was a gradient in the microbiological composition of the glacial ice samples, several CCA in which the distance to the volcanic focus was considered were carried out. These analyses demonstrated that there was a clear correlation between the distance to the volcano and the presence of some groups of bacteria (Figure 4A). Several phyla were associated with the volcanic focus (i.e., Thermotogae), while others were grouped away from the volcano (i.e., Betaproteobacteria). This same effect could be observed when microeukaryotes data were analyzed (Figure 4B).

Then, several CCA with all nutrients (NH_4_^+^, NO_2_^−^, NO_3_^−^, TDN, and DOC) were used to estimate the proportion of the community diversity attributable to each nutrient concentration. In these analyses, arrows indicate the direction in which phyla increase in abundance [44]. Arrows that point in the same direction indicate positive correlation, perpendicular arrows indicate lack of correlation and arrows pointing in the opposite direction indicate negative correlation. According to Figure 5, several phyla of bacteria were grouped along the arrows (Figure 5A), while microeukaryotes did not follow the direction of arrows so accurately (Figure 5B). The quantification of these results, expressed by the eigenvalues (Table 3), demonstrated that both bacterial and eukaryotic populations were dependent on these nutrients. Nevertheless, the eigenvalues explained the diversity for microeukaryotes (Table 3, analysis no. 5) better than they did for bacteria (analysis no. 5). 

When these CCA analyses were performed for the main chemical elements contained in the meltwater samples (Figure 6, Table 3), it could be shown that there was a correlation between the chemical composition of the glacial ice and the composition of the microbiomes along the volcanic gradient. According to eigenvalues, the correlation between the distribution of bacteria and the concentrations of these chemical elements was significant (Table 3, analysis no. 3). In contrast, microeukaryote populations were poorly correlated with the concentration of chemical elements (analysis no. 6). In conclusion, both volcanic chemical elements (i.e., Fe, Cd, S) and some bacterial groups (i.e., Caldithrix, Deferribacteres, Thermi, Thermodesulfobacteria, Thermotogae) follow the same volcanic gradient. It had already been reported that in Antarctica, the nutrient content in glacial ice is very low and volcanic ash contributes to its increase, favoring the massive development of some types of microorganisms [15].

Overall, since the diversity of bacteria was significantly correlated with the concentrations of both organic nutrients and chemical elements, it appears that bacteria were both autotrophs (photo and chemolithoautotrophs) and heterotrophs. However, eukaryotic populations were only correlated with organic nutrients, and therefore they must be photoautotrophs or heterotrophs.

### 3.5. Biogeochemical Cycles Inferred from Taxonomy

Given prokaryotes’ importance in biogeochemical transformations, a deeper knowledge of their involvement in polar biogeochemical cycles is required to understand their contributions to the wider ecosystem. In our glacier samples, some key members of carbon, nitrogen, phosphorus, iron, and sulfur cycles were identified (Figure 7, Appendix A).

Organic compounds in glacier ice must be biologically synthesized by CO_2_ fixation by phototrophs, i.e., the microeukaryotes *Thalassiosira antarctica* and *Hydrurus foetidus* identified in ROJ and MAC samples or chemolithotrophs such as *Streptomyces* [45,46]. In our research, we have also identified some aerobic CO oxidizers as *Corynebacterium* [47], especially in the eastern glaciers. Furthermore, organic compounds can be originated outside the glacier from animal, plant, or human remains. Afterward, this organic matter is degraded by considerable diversity of fungi such as *Chytridiaceae* sp., more abundant in the eastern islands. Several methylotrophs were identified in TRA and QUI glaciers in Greenwich Island (i.e., *Methylobacterium* and *Methylobacillus*), which can aerobically catabolize methane and many other C1 compounds. In the eastern glaciers, a large number of bacteria that metabolize other organic compounds were found, probably coming from human or animal contamination; among which Enterobacteriaceae stands out (i.e., *Enterobacter*, *Escherichia*). Some acetogens, which reduce carbonate to acetate, were also identified mainly in the eastern islands samples (*Acetobacterium*). Among heterotroph microeukaryotes, a high diversity of alveolates was found, especially ciliates and dinoflagellates. These microorganisms (i.e., *Paramecium*, *Gonyaulax*) often feed on bacteria or contain other endosymbiotic microorganisms that synthesize vitamins or other growth factors used by the host cell [32].

Moreover, organic matter deposited by marine birds, especially penguins, accumulates giving rise to the formation of ornithogenic soils where phosphatization is the main soil-forming process. This procedure greatly enhances nutrient availability, especially P and N. Some notable nesting zones are located in the South Shetland archipelago, especially on Deception and Barrientos islands. These two areas with ornithogenic soil are quite close to the ROJ and MAC glaciers in Deception Island [48], and QUI in Greenwich Island [27], sampling points 1, 2, and 6 respectively (Figure 7). In this type of soil, several bacterial species were identified, for example, *Clostridium frigoris* [48], the most abundant species in QUI samples (Appendix A). Some nitrogen-assimilating microorganisms have been found in the glacial ice samples analyzed, especially in the eastern islands, for instance, *Pseudomonas* or *Mycobacterium pinnipedii*, which has been described as an infectious agent in seals [49]. Nitrification in these glaciers is performed by bacteria as *Nitrosococcus* and *Nitrosovibrio* (ammonia-oxidation) followed by oxidation of nitrite to nitrate carried out by *Nitrobacter*. These aforementioned microorganisms (especially *Pseudomonas*) and some others (*Rhizobium*, *Enterobacter*), which are also frequent in the islands with penguins, participate very actively in the phosphorus cycle. Regarding microeukaryotes, the flagellate *Heteromita* was very abundant in all samples. This protist has been shown to affect the bacterial community and to excrete ammonia as a main form of nitrogen [50]. It has also been reported that diatoms in dark and anoxic environments might be involved in anaerobic nitrate respiration [51]. The most abundant eukaryotic genus in ECO samples was *Sporobolomyces* (Appendix A); this yeast produces a constitutive nitrate reductase and can use nitrate as the sole nitrogen source. The obtained nitrite is in turn reduced and assimilated by the yeast [52].

The microorganisms that participate in the sulfur cycle have been identified mainly in the vicinity of the volcano, forming a gradient of abundance that begins on Deception Island and goes east (Appendix A). Some of the most abundant species identified in this gradient were *Sulfobacillus yellowstonensis*, *Desulfosporosinus meridiei*, *Desulfurispora thermophila*, and *Desulfotomaculum thermoacetoxidans*. Additionally, several species of the *Thiomonas* and *Thiobacillus* genera were identified in the glaciers of Deception Island. These bacteria oxidize hydrogen sulfide to sulfur and sulfate, a key nutrient for microalgae, which enables the presence of these microeukaryotes and their predator protists in glacial ice. The bacterium, *Paracoccus*, identified in the eastern glaciers is capable of oxidizing reduced sulfur compounds. In general, the microorganisms that participate in the sulfur cycle are those that best fit the volcanic gradient. Appendix A represents the abundance of the main bacterial species participating in the sulfur cycle along the volcanic gradient. If these microorganisms involved in the sulfur cycle have indeed spread throughout the archipelago South Shetland as a result of the explosive volcanic eruptions is an appealing question to investigate. It can be observed that the abundance of these microorganisms is greater in the glaciers closest to the volcano and decreases with distance. Microorganisms along aerosol gradients have been widely studied [53]. Several emission sources and short- to long-distance dispersal have been identified. Among them, volcanoes have the capability to load the troposphere and stratosphere with ash and microorganisms and affect long-distance transport rates on a global scale. For example, *Bacillus luciferensis*, was firstly identified from a sample collected at an altitude of 20 km that had been dispersed by an explosive volcanic eruption [54]. This aerial transport has also been reported for other species of viable bacteria and fungi in several reports [55,56]. 

The iron cycle in glacier samples was represented by Fe^3+^ reducer bacteria as *Carboxydocella ferrireducens* or *Thermovenabulum ferriorganovorum*, which are also capable of reducing sulfite, thiosulfate, and elemental sulfur [57,58]. Some identified oxidizers of Fe^2+^ were *Ferrimicrobium* and *Gallionella ferruginea*. There are few eukaryotic microorganisms involved in the iron cycle, although some protists such as *Chlamydomonas* sp. are resistant to high concentrations of metals [59]. Other microorganisms that participate in unusual biogeochemical cycles typical of volcanic areas, for example, selenium metabolizing microorganisms, were also found in ice samples from Deception Island (i.e., *Selenomonas infelix*) [60].

## 4. Conclusions

Our research shows that (i) in the studied South Shetland glaciers, great diversity was observed when compared with microbiomes of other Antarctic glaciers or frozen deserts; (ii) the communities of microorganisms were isolated from small clusters within the ice. Even the compositions of the microbial populations identified in the glaciers from the same island were different; (iii) most of the 16S and 18S rRNA sequences searched in the databases were identified as “Unclassified” OTUs. There is still a huge work to be done in researching environmental microorganisms in polar environments; (iv) englacial microorganisms play an important role in the glacial microbiome since they participate in the biogeochemical cycles of the ecosystem; (v) there is a correlation between the chemical composition of the glacial ice and the composition of the microbial communities along the volcanic gradient. The bacteria that take part in the sulfur cycle are those that best fit the volcanic gradient; (vi) the eastern islands are those that suffer the greatest human pressure and their glaciers contain microorganisms typical of this contamination (Enterobacteriaceae, *Helicobacter*); (vii) some microorganisms, for example, photosynthesizers, raise the question of whether they live in the englacial zone or they have been dragged from the upper layers of the glacier. Future studies should check whether they are metabolically active and study their mechanisms of adaptation using transcriptomic and proteomic techniques; (viii) lastly, glaciers and especially those in volcanic regions such as Iceland or the Deception Island in Antarctica are being widely studied as models for astrobiological exploration; the results of this work can expand the settings in which life might be possible.

## Figures and Tables

**Figure 1 microorganisms-09-00392-f001:**
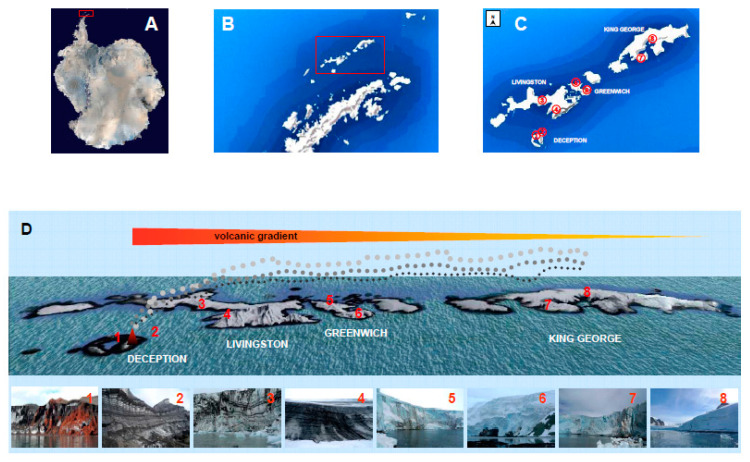
Location of sampling sites in South Shetland Islands, Antarctica. Satellite photographs of Antarctic continent (**A**) and Antarctic Peninsula (**B**) with South Shetland Islands boxed (ESA copyright-CC BY-SA IGO3.0; https://creativecommons.org/licenses/by-sa/3.0/igo/). (**C**) Map of Deception, Livingston, Greenwich, and King George Islands indicating the studied glaciers. (**D**) Volcanic gradient and setting of samples collected at glaciers: 1 Rojo; 2 Macaroni; 3 Johnson; 4 Hurd; 5 Quito; 6 Traub; 7 Ecology; 8 Machu Picchu. Ash and lava embedded in glacial ice can be observed along the gradient in the lower images.

**Figure 2 microorganisms-09-00392-f002:**
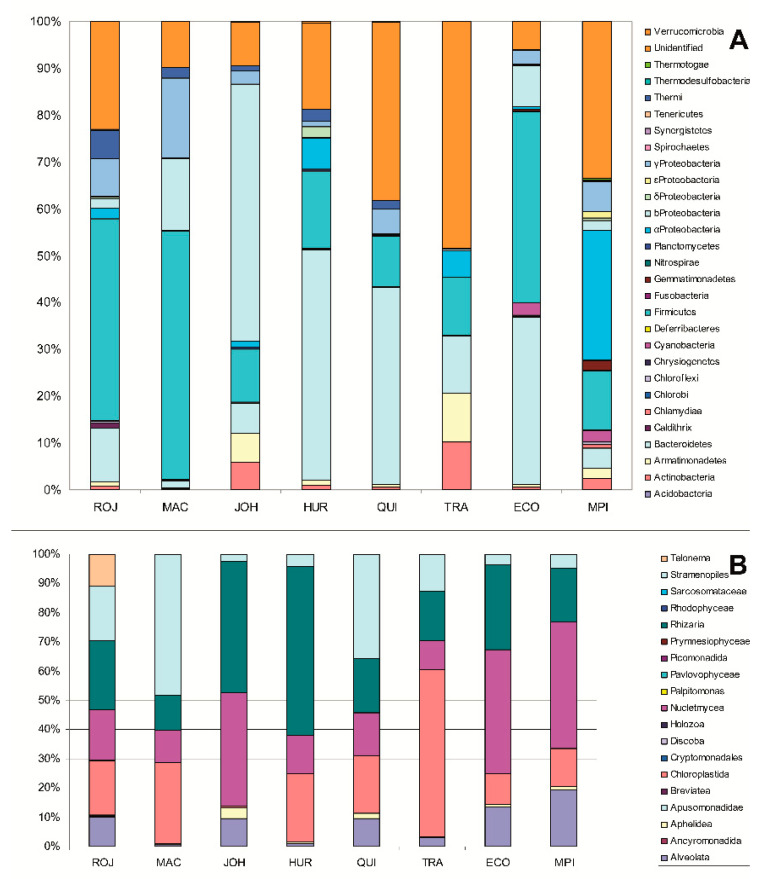
Composition of microbial communities. Relative abundances of major taxa of (**A**) bacteria and (**B**) microeukaryotes in the englacial ecosystem based on 16S and 18S rRNA gene sequencing.

**Figure 3 microorganisms-09-00392-f003:**
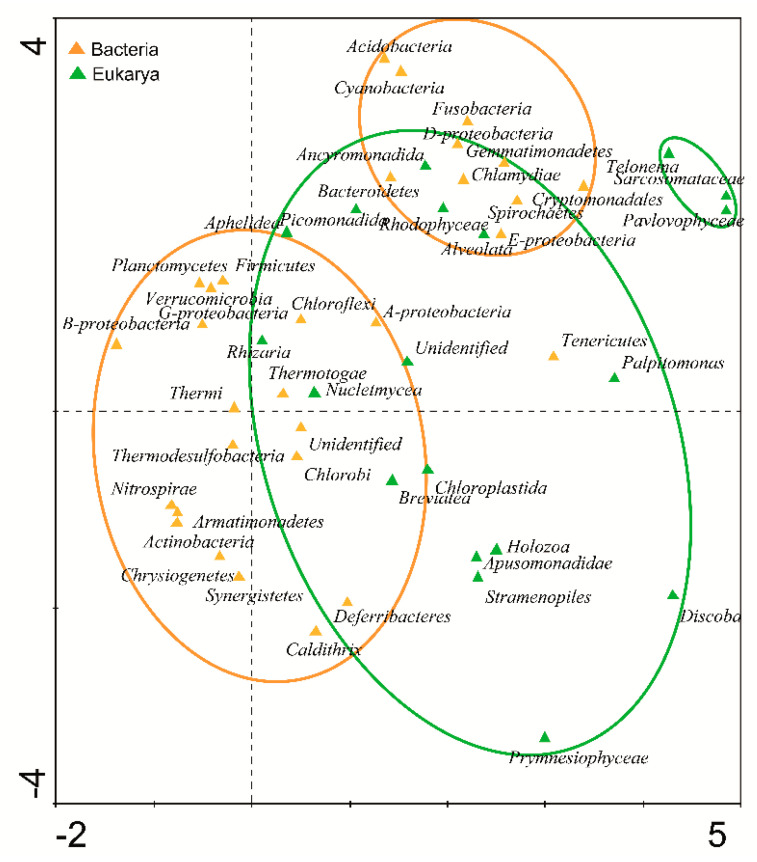
The relative abundances of the microbial phyla of bacteria and microeukaryotes were analyzed by detrended corresponded analysis (DCA). The diagram displays triangles that represent phyla of bacteria (orange) and microeukaryotes (green). Venn diagrams cluster and discriminate areas in which bacteria (orange diagrams) or microeukaryotes (green diagrams) are more abundant.

**Figure 4 microorganisms-09-00392-f004:**
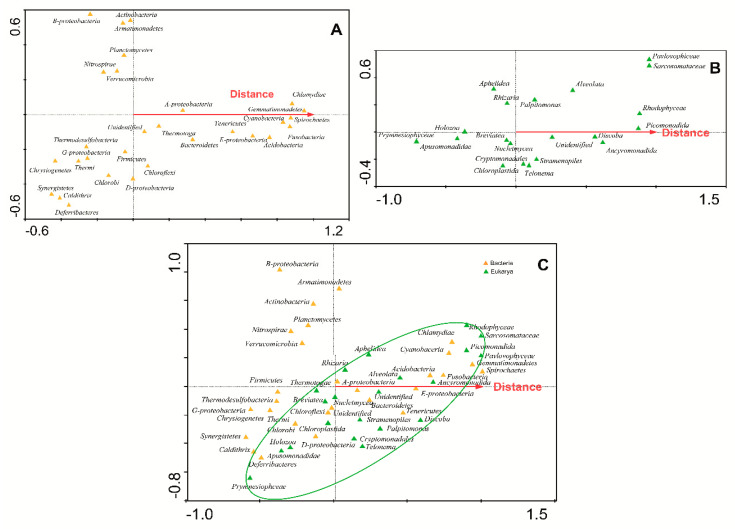
Canonical correspondence analysis (CCA) of the microbial phyla with respect to the distance from the volcano. CCA of phyla of bacteria (**A**) and microeukaryotes (**B**) identified in glaciers with respect to the distance from the volcano. The diagram displays triangles that represent taxonomic groups. (**C**) CCA of all the microbial phyla (bacteria and microeukaryotes) identified in glaciers with respect to the distance from the volcano. The diagram displays orange triangles for bacteria and green triangles for microeukaryotes. Venn diagram clusters an area in which microeukaryotes are more abundant.

**Figure 5 microorganisms-09-00392-f005:**
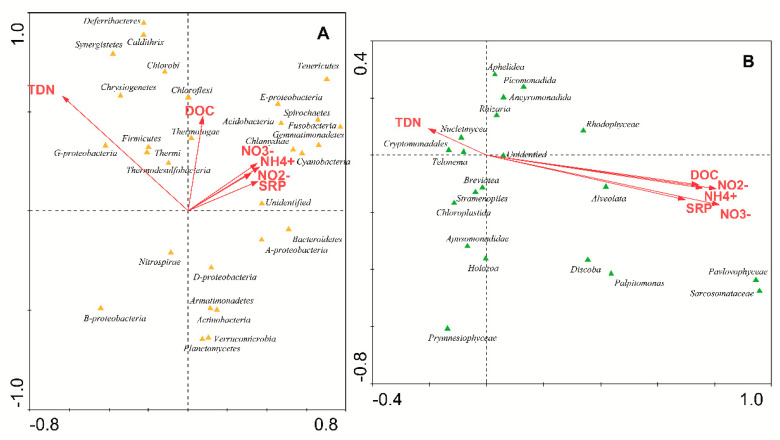
CCA of the microbial phyla of (**A**) bacteria and (**B**) microeukaryotes identified in glaciers with respect to the concentration of nutrients (NH_4_^+^, NO_2_^-^, NO_3_^-^, total dissolved nitrogen (TDN), soluble reactive phosphorous (SRP), and dissolved organic carbon (DOC)).

**Figure 6 microorganisms-09-00392-f006:**
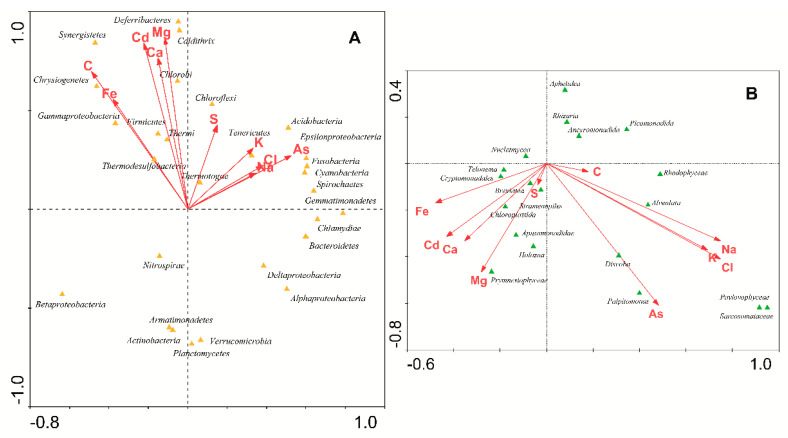
CCA of the microbial phyla of (**A**) bacteria and (**B**) microeukaryotes identified in glaciers with respect to the concentration of main elements (As, C, Ca, Cd, Cl, Fe, K, Mg, Na, S).

**Figure 7 microorganisms-09-00392-f007:**
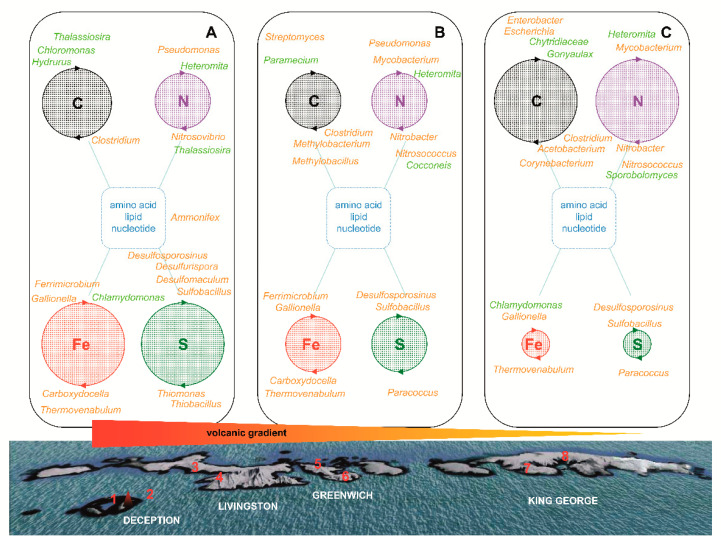
Insights into biogeochemical cycles based on the volcanic gradient along South Shetland glaciers. Some key members of carbon, nitrogen, iron, and sulfur cycles in glaciers were represented at the genus level (bacteria colored in orange and microeukaryotes colored in green). The size of the circles was calculated by the number of operational taxonomic units (OTUs) occurring at a frequency of more than 3% at sample points: (**A**) sample points 1 and 2, (**B**) sample points 3 to 6 and (**C**) sample points 7 and 8.

**Table 1 microorganisms-09-00392-t001:** Glacier coordinates.

ISLAND	Deception		Livingston		Greenwich		King George	
GLACIER	ROJ ^a^	MAC	JOH	HUR	QUI	TRA	ECO	MPI
GPS Coordinates	62°57′30.5′S, 60°35′55.6′’W	62°54′09.2′’S, 60°32′03.6′’W	62°38′60.0′’S, 60°21′00.0′’W	62°41′0.5”S, 60°20′74”W	62°27′00.0′’S, 59°45′00.0′’W	62°27′59.7”S, 59°45′23.8”W	62°09′60.0′’S, 58°28′00.0′’W	62°6′0”S, 58°28′0”W

^a^ Sample name: Rojo (ROJ), Macaroni (MAC), Johnson (JOH), Hurd (HUR), Quito (QUI), Traub (TRA), Ecology (ECO), Machu Pichu (MPI).

**Table 2 microorganisms-09-00392-t002:** Geochemical properties and chemical analysis of soluble nutrients in the meltwater.

ISLAND	Deception		Livingston		Greenwich		King George	
GLACIER	ROJ ^e^	MAC	JOH	HUR	QUI	TRA	ECO	MPI
DGV ^a^ (Km)	1.1	2.1	3.5	36.8	68	68.5	136	142
Tª (°C)	−0.5	0	−0.1	−3	−5	−5.1	−5.2	−5.3
pH	5.5	4.5	4.5	5.0	5.0	5.0	5.5	4.5
NH_4_^+^ (μM)	2.21 (0.09)	1.87 (0.25)	1.52 (0.14)	1.32 (0.08)	1.20 (0.25)	1.01 (0.18)	**60.04 (4.21)**	55.04 (2.31)
NO_2_^−^ (μM)	3.99 (0.29)	3.01 (0.29)	4.21 (0.23)	5.14 (0.21)	BD ^f^	BD	55.87 (5.06)	**59.11 (3.66)**
NO_3_^−^ (μM)	18.01 (0.99)	6.52 (0.44)	7.01 (0.11)	7.41 (0.17)	4.25 (0.24)	3.02 (0.35)	90.81 (6.33)	**99.87 (5.43)**
TDN ^b^	91.35 (0.12)	**188.04 (8.32)**	110.32 (6.33)	100.18 (7.54)	90.24 (3.22)	77.01 (4.28)	122.11 (6.55)	100.11 (2.55)
SRP ^c^	0.88 (0.14)	0.42 (0.12)	0.54 (0.14)	0.66 (0.01)	0.41 (0.12)	0.51 (0.01)	**1.38 (0.01)**	1.24 (0.01)
DOC ^d^	44.32 (5.21)	77.21 (8.45)	56.14 (5.34)	40.28 (8.22)	55.01 (8.32)	41.37 (6.37)	111.12 (5.22)	**120.22 (5.12)**

Concentrations are expressed in μM (± SEM) of three replicates. ^a^ DGV: Distance glacier-volcano; ^b^ TDN: Total dissolved nitrogen; ^c^ SRP: Soluble reactive phosphorus; ^d^ DOC: dissolved organic carbon; ^e^ Sample name: Rojo (ROJ), Macaroni (MAC), Johnson (JOH), Hurd (HUR), Quito (QUI), Traub (TRA), Ecology (ECO), Machu Pichu (MPI); ^f^ BD: below detection or <0.3 μM; the highest nutrient values are marked in bold.

**Table 3 microorganisms-09-00392-t003:** Summary of canonical correspondence analysis and correlations.

Type of Microorganism	No. of Analysis	Type of Analysis	Variables	λ1	λ2	λ3	λ4
Bacteria	1	CCA	DGV ^a^	0.715	0.000	0.000	0.000
	2	CCA	NH_4_^+^, NO_2_^−^, NO_3_^−^, TDN, DOC	0.794	0.708	0.795	0.440
	3	CCA	As, C, Ca, Cd, Cl, Fe, K, Mg, Na, S	0.859	0.752	0.795	0.814
Eukarya	4	CCA	DGV ^a^	0.743	0.000	0.000	0.000
	5	CCA	NH_4_^+^, NO_2_^-^, NO_3_^-^, TDN, DOC	0.982	0.905	0.888	0.681
	6	CCA	As, C, Ca, Cd, Cl, Fe, K, Mg, Na, S	0.202	0.054	0.040	0.017

^a^ DGV: Distance glacier-volcano.

## Data Availability

Sequences obtained from 16S and 18S rRNA sequencing were deposited in NCBI Short Read Archive (SRA).

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
