# Peer review of "Microbial Community Structure Driven by a Volcanic Gradient in Glaciers of the Antarctic Archipelago South Shetland"

_microorganisms, 2021, doi:10.3390/microorganisms9020392_

Round 1

Reviewer 1 Report

Here is the review of manuscript entitled " Microbial community structure driven by a volcanic gradient in glaciers of the Antarctic Archipelago South Shetland".

The object of the paper was to research composition of bacterial and microeukaryote communities in ice on eight glaciers in South Shetland Islands, Antarctica. Additionally, the taxonomic identification was performed by 16S rRNA and 18S rRNA high throughput sequencing. The authors tested whether there was a relationship between the chemical composition of the ice and the presence of microorganisms in each glacier. The results analyzed by statistical multivariate analysis showed that a significant correlation exists between the chemical composition of the glacial ice and the composition of the microbial communities along the volcanic gradient. This is especially evident in the microorganisms that are actively engaged in the sulfur cycle. On the eastern island, the human influence had a noticeable impact on the microbial communities.

The research methods used in the study are suitable, even though these are not very well explained. The authors mostly cited earlier works so the reader must refer to many papers to understand the methods employed in the paper. I suggest that authors extend methods part for sequencing work.

A list of proposed corrections in the text:

14 microeukariotes -> microeukaryotes
15 18S rRNA sequencing and -> 18S rRNA high throughput sequencing (HTS) and
22 16S/18S rRNA sequencing ->16S/18S rRNA high throughput sequencing
46 east the -> east to the
88 molecular (rRNA gene)-based approach -> molecular (high throughput rRNA gene sequencing)-based approach
112 the in meltwater -> in the meltwater
141-142 as previously reported [11, 12] (comment: Please describe the procedures used for amplification and sequencing here. You are referring to previous paper which slows the reading process. Also, it is not evident which primers you are using and which groups of microeukaryotes you are targeting, which is especially important to the reader)
144 grouped for their -> grouped according to their
146 than genus. -> than genus rank.
172 over many years -> over the many years
194 to the waves of the sea -> to the sea waves
195 3.2.16. S and 18S rRNA gene clone libraries. -> 3.2. 16S and 18S rRNA gene clone libraries
242-243 typical of this presence (comment: Presence of which? Human presence? Please make sentence more clear.)
320 taxonomy. -> taxonomy
399 identified OTUs "Unclassified" -> were identified as "Unclassified" OTUs
409 photosynthesizes -> photosynthesizers
411 check that they are -> check whether they are
436 References (comment: References are not numbered.)

Best,

Reviewer

Reviewer 2 Report

Major:

The abstract has to be rewritten, to describe clearly and precisely what results have the authors found.

Introduction section: too much information on glaciers instead of information on investigated microbial communities and the unique or ubiquitous of these microorganisms, which are the emphasis of this work.

Please describe Table 1 in one section instead of presenting the data separately in M&M and section 3.1.

The chemical gradients in the analyzed volcanos are sophisticated and statements are not fully based on the data (e.g. Cd, Cl), and why Cl was not detected in many sampling sites?

Conclusions: This work reported the microbial communities in the Antarctic Archipelago South Shetland by sequencing, most of the statements are highly speculative, as stated by the authors a large proportion of microorganisms are unclassified. The conclusions have to be revised and justified based on the experimental data.

Others:

Line 15, please replace concluded with showed

Line 16, please specify what kind of gradient has the authors detected/found in the microbial communities from west to east in the volcano.

Line 21, what kind of human pressure?

Line 28, have been recently described

Line 128, please correct NH4+

Line 170-172, move to Introduction.

Line 172-173, move to M&M.

Line 195, please separate 3.2. and 16 S….

Line 209, Figure 2 is not readable.

Line 212, rendered

Line 220, ? two subtitles?

Line 284, are there indications from the sequencing and microbial community analysis?

Line 290-292, please provide details of the statements, what kind of correlations one can conclude from the chemical elements along the volcano gradient?

Line 357-359, experimental results or only speculations?

Round 2

Reviewer 1 Report

Dear authors & the editor,

All requested changes have been included in version 2 of the manuscript, so I find it suitable for publication in Microorganisms journal.

Best,

Reviewer